# GPAvatar: Generalizable and Precise Head Avatar from Image(s)

**Xuangeng Chu**[1,2*]  **Yu Li**[2†]  **Ailing Zeng**[2]  **Tianyu Yang**[2]  **Lijian Lin**[2]  **Yunfei Liu**[2]
**Tatsuya Harada**[1,3†]
[1]The University of Tokyo   [2]International Digital Economy Academy (IDEA)   [3]RIKEN AIP
{xuangeng.chu, harada}@mi.t.u-tokyo.ac.jp
{liyu, zengailing, yangtianyu, linlijian, liuyunfei}@idea.edu.cn

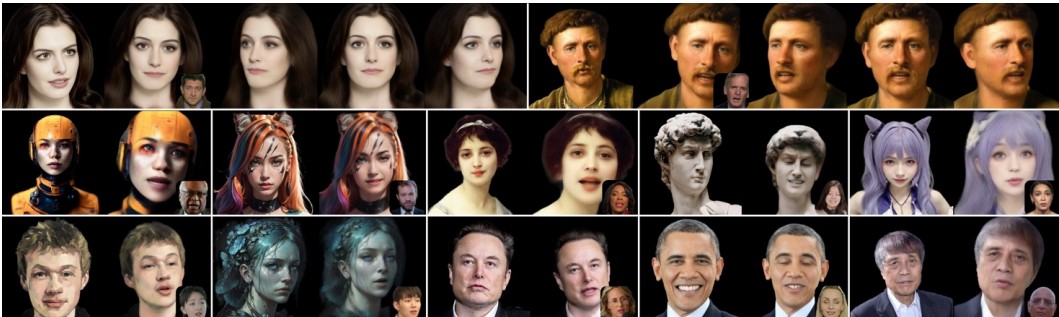

Figure 1: Our GPAvatar is able to reconstruct 3D head avatars from even a single input (*i.e.*, one-shot), with strong generalization and precise expression control. The leftmost images are the inputs, and the subsequent images depict reenactment results. Inset images display the corresponding driving faces. Additionally, the first row shows three novel view results.

## Abstract

Head avatar reconstruction, crucial for applications in virtual reality, online meetings, gaming, and film industries, has garnered substantial attention within the computer vision community. The fundamental objective of this field is to faithfully recreate the head avatar and precisely control expressions and postures. Existing methods, categorized into 2D-based warping, mesh-based, and neural rendering approaches, present challenges in maintaining multi-view consistency, incorporating non-facial information, and generalizing to new identities. In this paper, we propose a framework named GPAvatar that reconstructs 3D head avatars from one or several images in a single forward pass. The key idea of this work is to introduce a dynamic point-based expression field driven by a point cloud to precisely and effectively capture expressions. Furthermore, we use a Multi Tri-planes Attention (MTA) fusion module in the tri-planes canonical field to leverage information from multiple input images. The proposed method achieves faithful identity reconstruction, precise expression control, and multi-view consistency, demonstrating promising results for free-viewpoint rendering and novel view synthesis. Code and demos are available at https://xg-chu.github.io/project_gpavatar.

## 1 Introduction

Head avatar reconstruction holds immense potential in various applications, including virtual reality, online meetings, gaming, and the film industry. In recent years, this field has garnered significant attention within the computer vision community. The primary objective of head avatar reconstruction is to faithfully recreate the source head while enabling precise control over expressions and posture. This capability will facilitate the generation of desired new expressions and poses for the source portrait (Li et al., 2023a; Yu et al., 2023b; Li et al., 2023b).

---

*This work was partially done during an internship at IDEA.
†Corresponding Author.

Some exploratory methods have partially achieved this goal and can be roughly categorized into three types: 2D-based warping methods (Yin et al., 2022), mesh-based methods (Khakhulin et al., 2022), and neural rendering methods (Sun et al., 2023; Ma et al., 2023; Li et al., 2023a; Yu et al., 2023b; Li et al., 2023b). Among these, 2D-based methods warp the original image to new expressions with a warping field estimated from sparse landmarks, and then synthesize the appearance through an encoder and decoder. However, these methods struggle to maintain multi-view consistency when there are significant changes in head pose due to their lack of necessary 3D constraints. Furthermore, these methods are unable to effectively decouple expressions and identity from the source portrait, leading to unfaithful driving results. Mesh-based methods explicitly model the source portrait with a 3D Morphable Model (3DMM) (Blanz & Vetter, 1999; Paysan et al., 2009; Li et al., 2017; Gerig et al., 2018). By incorporating 3D information, these methods effectively address the issue of multi-view consistency. However, due to the limitations in the modeling and expressive capacity of 3DMM, the reconstructed head often lacks non-facial information such as hair, and the expressions are often unnatural. With the outstanding performance of NeRF (neural radiance field) in multi-view image synthesis, the latest methods have started to leverage NeRF for head avatar reconstruction(Xu et al., 2023; Zielonka et al., 2023; Zheng et al., 2022; Sun et al., 2023; Ma et al., 2023). Compared to 2D and mesh-based methods, NeRF-based methods have shown the ability to synthesize results that are 3D-consistent and include non-facial information. However, these methods can't generalize well to new identities. Some of these methods require a large amount of portrait data for reconstruction, and some involve time-consuming optimization processes during inference.

In this paper, we present a framework for reconstructing the source portrait in a single forward pass. Given one or several unseen images, our method reconstructs an animatable implicit head avatar representation. Some examples are shown in Fig. 1. The core challenge lies in faithfully reconstructing the head avatar from a single image and achieving precise control over expressions. To address this issue, we introduce a point cloud-driven dynamic expression field to precisely capture expressions and use a Multi Tri-planes Attention (MTA) module in the tri-planes canonical field to leverage information from multiple input images. The 3DMM point cloud-driven field provides natural and precise expression control and facilitates identity-expression decoupling. The merged tri-planes encapsulate a feature space that includes faithful identity information from the source portrait while modeling parts not covered by the 3DMM, such as shoulders and hair. The experiment verifies that our method generalizes well to unseen identities and enables precise expression control without test-time optimization, thereby enabling tasks such as free novel view synthesis and reenactment.

The major contributions of our work are as follows:

- We introduce a 3D head avatar reconstruction framework that achieves faithful reconstruction in a single forward pass and generalizes well to in-the-wild images.
- We propose a dynamic Point-based Expression Field (PEF) that allows for precise and natural cross-identity expression control.
- We propose a Multi Tri-planes Attention (MTA) fusion module to accept an arbitrary number of input images. It enables the incorporation of more information during inference, particularly beneficial for extreme inputs like closed eyes and occlusions.

## 2 RELATED WORK

### 2.1 TALKING HEAD SYNTHESIS

Previous methods for head synthesis can be categorized into deformation-based, mesh-based, and NeRF-based methods. Warping-based methods(Siarohin et al., 2019; Zakharov et al., 2020; Wang et al., 2021a; Yin et al., 2022; Drobyshev et al., 2022; Zhang et al., 2023) are popular among 2D generative methods. Usually, these methods apply deformation operations to the source image to drive the motion in the target image. Due to a lack of clear understanding and modeling of the 3D geometry of the head avatar, these methods often produce unrealistic distortions when the poses and expressions change a lot. Many subsequent works(Ren et al., 2021; Yin et al., 2022; Zhang et al., 2023) alleviated this problem by introducing 3DMM(Blanz & Vetter, 1999; Paysan et al., 2009; Li et al., 2017; Gerig et al., 2018), but this problem still exists and limits the performance of 2D methods. To completely address this problem, many 3DMM-based works(Feng et al., 2021; Danecek et al., 2022; Khakhulin et al., 2022) reconstruct animatable avatars by estimating 3DMM

parameters from portrait images. Among them, ROME(Khakhulin et al., 2022) estimates the 3DMM parameters, the offset of mesh vertex and the texture to render the results. However, although 3DMM provides strong priors for understanding the face, it focuses only on facial regions and cannot capture other detailed features such as hairstyles and accessories, and the fidelity is limited by the resolution of meshes, resulting in unnatural appearances in the reenactment images.

NeRF(Mildenhall et al., 2020) is a type of implicit 3D scene representation method known for its excellent performance in static scene reconstruction. Many works(Park et al., 2021a;b; Tretschk et al., 2021) try to extend it from static scenes to dynamic scenes, and there are also many works(Gafni et al., 2021; Zheng et al., 2022; Xu et al., 2023; Zielonka et al., 2023; Athar et al., 2023) that apply NeRF to human portrait reconstruction and animation. One of the research directions is to generate controllable 3D head avatars from random noise (Sun et al., 2023; Ma et al., 2023). While these methods can produce realistic and controllable results, achieving reconstruction requires GAN inversion, which is impractical in real-time scenarios. Another research direction is to utilize data from specific individuals for reconstruction (Gafni et al., 2021; Athar et al., 2022; Zheng et al., 2022; Xu et al., 2023; Bai et al., 2023; Zielonka et al., 2023). While the results are impressive, they cannot learn networks for different identities and require thousands of frames of personal image data, raising privacy concerns. At the same time, there are also some methods to use audio drivers to control avatar (Tang et al., 2022; Guo et al., 2021; Yu et al., 2023a), providing users with a more flexible and easy-to-use driver method.

## 2.2 ONE-SHOT HEAD AVATARS

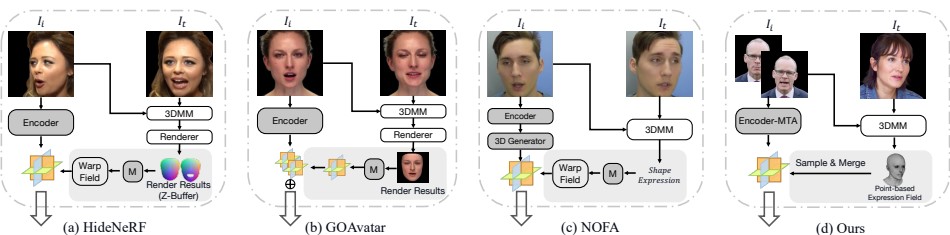

Figure 2: Differences from existing state-of-the-art methods. Existing methods may over-process expression information or use expression features, leading to expression detail loss. Our approach avoids this loss with a point-based expression field, and our method flexibly accepts single or multiple images as input, enhancing information gathering through our multi-tri-planes attention module.

To address these issues, some works(Trevithick et al., 2023; Hong et al., 2022; Li et al., 2023a;b; Yu et al., 2023b) have focused on reconstructing 3D avatars from arbitrary input images. Some methods can achieve static reconstruction(Trevithick et al., 2023; Hong et al., 2022), but they are unable to reanimate these digital avatars. There are also methods that utilize NeRF to achieve animatable one-shot forward reconstruction of target avatars, such as GOAvatar(Li et al., 2023b), NOFA(Yu et al., 2023b), and HideNeRF(Li et al., 2023a). GOAvatar (Li et al., 2023b) utilizes three sets of tri-planes to respectively represent the standard pose, image details, and expression. It also employs a fine-tuned GFPGAN (Wang et al., 2021b) network to enhance the details of the results. NOFA (Yu et al., 2023b) utilizes the rich 3D-consistent generative prior of 3D GAN to synthesize neural volumes of different faces and employs deformation fields to model facial dynamics. HideNeRF (Li et al., 2023a) utilizes a multi-resolution tri-planes representation and a 3DMM-based deformation field to generate reenactment images while enhancing identity consistency during the generation process. While these methods produce impressive results, they still have some limitations in expression-driven tasks. Some of these methods either rely on the rendering results of 3DMM as input to control deformation fields or directly use 3DMM parameters for expression-driven tasks. In such encoding and decoding processes, subtle facial expression information may inevitably be lost.

In this paper, we utilize the FLAME (Li et al., 2017) point cloud as prior and propose a novel 3D head neural avatar framework. It not only generalizes to unseen identities but also offers precise control over expression details during reenact and surpasses all previous works in reenactment image quality. Fig. 2 illustrates the differences between our method and existing approaches.

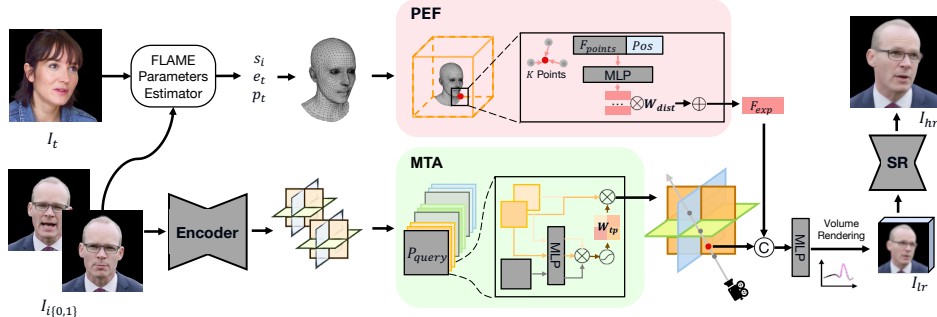

Figure 3: Overview: Our method mainly comprises two branches: one that captures fine-grained expressions with PEF (Sec. 3.2) and another that integrates information from multiple inputs through MTA (Sec. 3.1, 3.3). Finally, there is the rendering and super-resolution component (Sec. 3.4).

## 3 METHOD

In this section, we will describe our method. Our approach has the capability to faithfully reconstruct head avatars from any number of inputs and achieve precise reenactment. The overall process can be summarized as follows:

$$I_t = R(MTA(E(I_i)), PEF(\text{FLAME}(s_i, e_t, p_t), \theta), p_{cam}),$$

where $I_i$ represents the input image(s), $s_i$ is the shape parameter of the source image $I_i$, $e_t$ and $p_t$ is the desired expression and pose parameter. The canonical feature space is constructed by $E(I_i)$, and our Point-based Expression Field ($PEF$) is built with point cloud FLAME($s_i, e_t, p_t$) and point feature $\theta$. If there are multiple input images, their canonical feature space will be merged by the Multi-Tri-planes Attention module ($MTA$). Finally, $R$ is the volume rendering function that renders the reenactment image $I_t$ based on camera pose $p_{cam}$. The overall process is illustrated in Fig. 3. In the following, we will describe the canonical encoder in Sec. 3.1, explain how PEF controls expressions in Sec. 3.2, introduce how we fuse multiple inputs through MTA in Sec 3.3, discuss the rendering and super-resolution process in Sec. 3.4, and finally, we describe the training targets in Sec. 3.5.

### 3.1 CANONICAL FEATURE ENCODER

Due to the fact that the tri-planes representation has strong 3D geometric priors and strikes a good balance between synthesis quality and speed, we employ the tri-planes representation as our standard feature space. Specifically, inspired by GFPGAN (Wang et al., 2021b), our encoder follows a UNet structure, and during its up-sampling process, we use a StyleGAN structure. We generally keep the same setting as GFPGAN, except that our encoder maps the original image from $3\times512\times512$ to $3\times32\times256\times256$ to build a tri-planes feature space. We only modified the input and output layers to achieve this. In the experiment, we observed that this structure can effectively integrate global information from the input image during the down-sampling process, and then generate mutually correlated planes during the up-sampling process. In order to enhance the robustness of the encoder and adapt to arbitrary real-world inputs, we applied affine transformations to align the input 2D images using estimated head poses. Since we utilize a separate expression feature field encoded with PEF, the canonical feature space here lacks complete semantics on its own. Therefore, while many works based on tri-planes restrict the canonical feature space to have neutral expressions, here we do not impose any restrictions and train the encoder from scratch in an end-to-end manner.

### 3.2 POINT-BASED EXPRESSION FIELD

In this section, we will introduce how to build a controllable expression field based on point clouds. Many methods for head avatars rely on 3DMM parameters or rendering images to generate expressions. However, they either have limited expressive capabilities when directly using 3DMM parameters or lose details due to excessive encoding and decoding processes. Inspired by Point-NeRF (Xu et al., 2022), we directly use the point cloud from 3DMM to construct a point-based expression field, thereby avoiding over-processing and retaining expression details as much as possible.

Unlike attempts to reconstruct static scenes, our point-based expression field (PEF) aims to model dynamic expressions. To achieve this goal, we bind learnable weights to each FLAME vertex in the PEF. Due to the stable semantics and geometric topology of FLAME vertices, such as points representing the eyes and mouth not undergoing semantic changes across various expressions and identities, each neural point in the PEF also holds stable semantics and can be shared across different identities. During sampling features from PEF, we sample several nearest points to calculate the final feature for the sample position. If we sample the nearest points from the local region following (Xu et al., 2022), we may encounter limitations in representation capabilities, such that non-FLAME areas are modeled only in canonical feature space and parts related to expressions such as hair that are not included in FLAME may become fully rigid, making certain expressions unnatural. Therefore, we instead search for neighboring points in the entire space and use relative position encoding to provide the model with direction and distance information. Our approach liberates the representation capabilities of point features, and experiments have also confirmed that our method performs better. To achieve the synergy between the canonical feature space and PEF and harness the prior capabilities of point clouds, we remove the global pose from the FLAME pose and instead model it using the camera pose. This ensures that the point cloud is always in a canonical position near the origin. Since we sample features from both feature spaces, the semantic information and 3D priors from the PEF can also undergo collaborative learning with the canonical feature space.

The overall process of our PEF is as follows: for any given query 3D position $x$ during the NeRF sampling process, we retrieve its nearest $K$ points and obtain their corresponding features $f_i$ and positions $p_i$. Then, we employ linear layers to regress the features for each point, and finally combine these features based on positional weights, as shown in Eq. 1:

$$f_{exp,x} = \sum_i^K \frac{w_i}{\sum_j^K w_j} L_p(f_i, F_{pos}(p_i - x)), \text{where } w_i = \frac{1}{p_i - x}, \tag{1}$$

where $L_p$ is the linear layers and $F_{pos}$ is the frequency positional encoding function. During this process, the position of point $p_i$ changes as the FLAME expression parameters change, creating a dynamic expression feature field. This allows the FLAME to directly contribute to the NeRF feature space, avoiding the loss of information introduced by excessive processing. Due to the decoupling of the canonical tri-planes and PEF, we only create the canonical tri-planes once during inference, and the speed of PEF will affect the inference speed. Thanks to the efficient parallel nearest neighbor query, the PEF process can be completed quickly, greatly improving the speed of inference.

### 3.3 MULTI TRI-PLANES ATTENTION

Based on the aforementioned modules, we can obtain animatable high-fidelity results. However, since the source image can be arbitrary, this introduces some challenging scenarios. For example, there may be occlusions in the source image, or the eyes in the source image may be closed while the desired expression requires open eyes. In this situation, the model may generate illusions based on statistically average eye and facial features, but these illusions may be incorrect. Although this image cannot produce the truth about missing parts, we may have other images that supplement the missing parts. To achieve this goal, we have implemented an attention-based module to fuse the tri-planes features of multiple images, which is called Multi Tri-planes Attention (MTA).

Our MTA uses a learnable tri-plane to query multiple tri-planes from different images, generating weights for feature fusion, as shown in Eq. 2:

$$P = \sum_i^N \frac{w_i}{\sum_j^N w_j} E(I_i), \text{where } w_i = L_q(Q)L_k(E(I_i)), \tag{2}$$

where $I_i$ is the input image, $N$ is the number of input images, $E$ is the canonical encoder, $L_q$ and $L_k$ are the linear layers to generate queries and keys, and $Q$ is the learnable query tri-planes.

Through our experiments, we have demonstrated that our MTA effectively enhances performance and completes missing information in one-shot inputs, such as pupil information and the other half of the face in extreme pose variations. During training, we use two random frames as input and one frame as the target, while during inference, our MTA can accept any number of images as input. Furthermore, experiments show that our MTA can consistently fuse multiple tri-planes features from images of the same person captured at different times. Even when dealing with images of different individuals and styles, MTA can still produce reasonable results, showcasing its strong robustness.

### 3.4 Volume Rendering and Super Resolution

Given the camera's intrinsic and extrinsic parameters, we sample the rays and perform two-pass hierarchical sampling along these rays, followed by volume rendering to obtain 2D results. Due to the extensive computational resources required for high-resolution volume rendering, training and testing on high resolution become time-consuming and costly. A popular solution to this problem is the lightweight super-resolution module. In our work, we render low-resolution images at 128x128 resolution, and these low-resolution images consist of a 32-dimensional feature map with the first three dimensions corresponding to RGB pixel values. The super-resolution module we use is similar to our canonical feature space encoder, and like the encoder, we train this super-resolution module from scratch in an end-to-end manner.

### 3.5 Training Strategy and Loss Functions

We train our model from scratch using an end-to-end training approach. By sampling original and target images from the same video, we construct pairs of images with the same identity but different expressions and poses. During the training process, our primary objective is to make the reenactment images consistent with the target images. We use $L_1$ and perceptual loss (Johnson et al., 2016; Zhang et al., 2018) on both low-resolution and high-resolution reenactment images to achieve this objective, as shown in the Eq. 3:

$$\mathcal{L}_{rec} = ||I_{lr} - I_t|| + ||I_{hr} - I_t|| + \lambda_p(||\varphi(I_{lr}) - \varphi(I_t)|| + ||\varphi(I_{hr}) - \varphi(I_t)||), \tag{3}$$

where $I_t$ is the reenactment target image, $I_{lr}$ and $I_{hr}$ are the low-resolution and high-resolution reenactment results, $\varphi$ is the AlexNet (Krizhevsky et al., 2012) used in the perceptual loss, and $\lambda_p$ is the weight for the perceptual loss.

Additionally, we add a density-based norm loss as shown in the Eq. 4:

$$\mathcal{L}_{norm} = ||d_n||_2, \tag{4}$$

where $d_n$ is the density used in volume rendering (Mildenhall et al., 2020). This loss encourages the total density of NeRF to be as low as possible, thereby encouraging reconstructions that closely adhere to the actual 3D shape and avoid the appearance of artifacts. The overall training objective is as:

$$\mathcal{L}_{overall} = \lambda_r \mathcal{L}_{rec} + \lambda_n \mathcal{L}_{norm}, \tag{5}$$

where $\lambda_r$ and $\lambda_n$ are the weights that balance the loss.

## 4 Experiments

In this section, we will first introduce the dataset we use, the implementation details of our method, and the baselines of our work. We will then compare our method with existing approaches using a variety of metrics.

### 4.1 Experiment Setting

**Datasets.** We use the VFHQ (Xie et al., 2022) dataset to train our model. This dataset comprises clips from various interview scenarios, and we utilized a subset consisting of 8,013 video clips. From these videos, we extracted 240,390 frames to create our training dataset. During the training process, we randomly sampled frames from the same video to create pairs of images with the same identity but different expressions. One frame was used as the target for reenactment, while the others served as source images. Given that our method can accept any number of inputs, in each iteration, we sampled two inputs with a 70% probability and one input with a 30% probability. Regarding evaluation, we assessed our method on the VFHQ dataset (Xie et al., 2022) and the HDTF dataset (Zhang et al., 2021). It's important to note that our model was not fine-tuned on the HDTF dataset. In the evaluation process, we used the first frame of each video as the source image, with the remaining frames as targets for reenactment.

**Evaluation Metrics.** We evaluated all methods on both same-identity and cross-identity reenactment tasks. For the cross-identity reenactment task, due to the lack of ground truth, we evaluated the

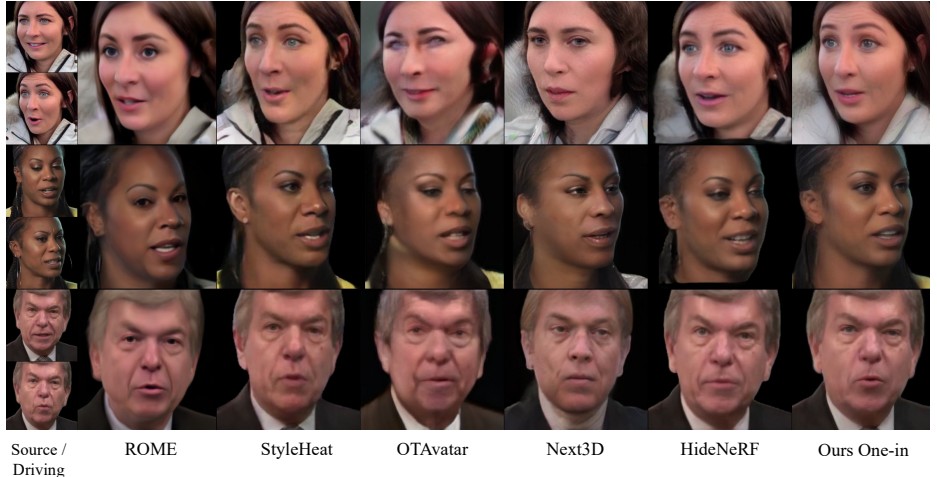

Source /    ROME        StyleHeat    OTAvatar     Next3D      HideNeRF    Ours One-in
Driving

Figure 4: Qualitative results on VFHQ (Xie et al., 2022) and HDTF (Zhang et al., 2021) datasets. The first two rows are from VFHQ and the third row is from HDTF.

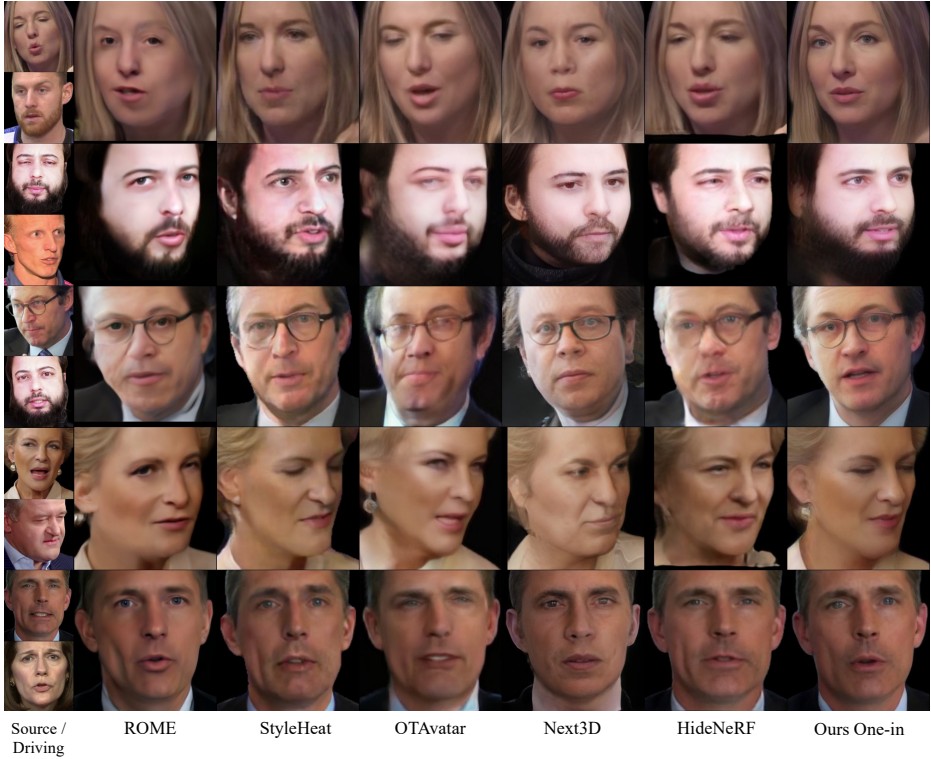

Source /    ROME        StyleHeat    OTAvatar     Next3D      HideNeRF    Ours One-in
Driving

Figure 5: Qualitative results on VFHQ (Xie et al., 2022) and HDTF (Zhang et al., 2021) datasets. The first four rows are from VFHQ and the last row is from HDTF.

cosine similarity of identity embeddings (CSIM) based on ArcFace (Deng et al., 2019) between the reenacted frames and source images to assess identity consistency during the reenactment task. We also used the Average Expression Distance (AED) and Average Pose Distance (APD) metrics based on (Danecek et al., 2022) to assess the accuracy of expression and pose driving. In the same-identity reenactment task, where ground truth frames are available, in addition to the aforementioned metrics, we evaluate PSNR, SSIM, L1, and LPIPS metrics between the reenacted frames and ground truth frames. We also calculated the Average Key-point Distance (AKD) based on (Bulat & Tzimiropoulos, 2017) as another reference for expression reenactment accuracy.

**Implementation Details.** Our framework is built upon the PyTorch framework (Paszke et al., 2017), and during the training process, we employ the ADAM (Kingma & Ba, 2014) optimizer with a learn-

Table 1: Quantitative results on the VFHQ (Xie et al., 2022) dataset. For a fair comparison, we compare one-shot results using the first frame input. Ours Two-in uses both the first and last frames. Entries in green are the best ones in a one-shot setting.

| Method | Self Reenactment | | | | | | | | Cross-Id Reenactment | | |
|---|---|---|---|---|---|---|---|---|---|---|---|
| | PSNR↑ | SSIM↑ | LPIPS↓ | CSIM↑ | L1↓ | AED↓ | APD↓ | AKD↓ | CSIM↑ | AED↓ | APD↓ |
| ROME (Khakhulin et al., 2022) | 19.88 | 0.735 | 0.237 | 0.679 | 0.060 | 0.497 | 0.017 | 4.53 | 0.531 | 0.936 | 0.026 |
| StyleHeat (Yin et al., 2022) | 19.95 | 0.738 | 0.251 | 0.603 | 0.065 | 0.593 | 0.024 | 5.30 | 0.506 | 0.961 | 0.038 |
| OTAvatar (Ma et al., 2023) | 18.10 | 0.600 | 0.346 | 0.660 | 0.092 | 0.734 | 0.035 | 6.05 | 0.514 | 0.962 | 0.059 |
| Next3D (Sun et al., 2023) | 19.95 | 0.656 | 0.281 | 0.631 | 0.066 | 0.727 | 0.026 | 5.17 | 0.482 | 0.996 | 0.036 |
| HideNeRF (Li et al., 2023a) | 20.07 | 0.745 | 0.204 | 0.794 | 0.056 | 0.521 | 0.031 | 5.33 | 0.558 | 1.024 | 0.044 |
| Ours One-in | 22.08 | 0.765 | 0.177 | 0.789 | 0.039 | 0.434 | 0.017 | 3.53 | 0.558 | 0.910 | 0.034 |
| Ours Two-in | 22.86 | 0.779 | 0.169 | 0.771 | 0.035 | 0.411 | 0.017 | 3.44 | 0.551 | 0.907 | 0.034 |

Table 2: Quantitative results on the HDTF (Zhang et al., 2021) dataset. For a fair comparison, we compare one-shot results using the first frame input. Ours Two-in uses both the first and last frames. Entries in green are the best ones in a one-shot setting.

| Method | Self Reenactment | | | | | | | | Cross-Id Reenactment | | |
|---|---|---|---|---|---|---|---|---|---|---|---|
| | PSNR↑ | SSIM↑ | LPIPS↓ | CSIM↑ | L1↓ | AED↓ | APD↓ | AKD↓ | CSIM↑ | AED↓ | APD↓ |
| ROME (Khakhulin et al., 2022) | 20.84 | 0.722 | 0.176 | 0.781 | 0.044 | 0.540 | 0.012 | 3.93 | 0.721 | 0.929 | 0.017 |
| StyleHeat (Yin et al., 2022) | 21.91 | 0.772 | 0.210 | 0.705 | 0.045 | 0.527 | 0.015 | 3.69 | 0.666 | 0.902 | 0.027 |
| OTAvatar (Ma et al., 2023) | 20.50 | 0.695 | 0.241 | 0.765 | 0.064 | 0.681 | 0.020 | 5.15 | 0.699 | 1.047 | 0.034 |
| Next3D (Sun et al., 2023) | 20.35 | 0.723 | 0.217 | 0.730 | 0.048 | 0.644 | 0.022 | 4.19 | 0.622 | 1.014 | 0.026 |
| HideNeRF (Li et al., 2023a) | 21.38 | 0.803 | 0.147 | 0.907 | 0.038 | 0.499 | 0.027 | 4.33 | 0.803 | 1.031 | 0.032 |
| Ours One-in | 24.21 | 0.834 | 0.131 | 0.871 | 0.029 | 0.427 | 0.012 | 3.06 | 0.790 | 0.869 | 0.020 |
| Ours Two-in | 25.36 | 0.849 | 0.122 | 0.851 | 0.026 | 0.406 | 0.012 | 3.01 | 0.769 | 0.837 | 0.021 |

ing rate of 1.0e-4. We conducted training on 2 NVIDIA Tesla A100 GPUs, with a total batch size of 8. During the training process, our PEF searches for the nearest $K=8$ points, while MTA selects two frames as source images. Our approach employs an end-to-end training methodology. The training process consists of 150,000 iterations and the full training process consumes approximately 50 GPU hours, showing its resource utilization efficiency. During the inference time, our method achieves 15 FPS when running on an A100 GPU. More details can be found in the supplementary materials.

## 4.2 MAIN RESULTS

**Baseline Methods.** We compared our method with five state-of-the-art existing methods, including StyleHeat (Yin et al., 2022) (2D-based warping), ROME (Khakhulin et al., 2022) (mesh-based), OTAvatar (Ma et al., 2023), Next3D (Sun et al., 2023; Roich et al., 2021) (based on NeRF and 3D generative models), and HideNeRF (Li et al., 2023a), which is most similar to our setup. All results were evaluated using official code implementations.

**Self-Reenactment Results.** We begin by evaluating the synthesis performance when the source and driving image are the same person. Tab. 1 and Tab. 2 show the quantitative results on VFHQ and HDTF, respectively. Notably, our approach exhibits a significant advantage over other state-of-the-art methods in terms of both synthesis quality metrics (PSNR, SSIM, LPIPS, and L1) and expression control quality metrics (AED and AKD). Qualitative results on VFHQ and HDTF are visually demonstrated in Fig. 4. These results showcase that our method not only excels in synthesis quality but also captures subtle expressions, as exemplified by the surprised expression in the first row and the angry expression in the second row. Importantly, our model achieved these results without any training or fine-tuning on the HDTF dataset, thus demonstrating the robust generalization capability of our approach.

**Cross-Identity Reenactment Results.** We also evaluate the synthesis performance when the source and the driving images contain different persons. Tab. 1 and Tab. 2 show quantitative results, and Fig. 5 showcases qualitative results. Due to the absence of ground truth data, a quantitative evaluation of synthesis performance is not feasible, but the qualitative results evident that our method excels in expression control. These results show the efficacy of our approach in scenarios where the source and driving images are from different individuals.

**Multiple images input.** In addition to quantitative results, we further illustrate the advantages of multi-input methods in challenging scenarios, as shown in Fig. 6, such as closed eyes and significant pose variations. The results demonstrate that employing multiple inputs can further enhance synthesis quality while maintaining precise expression control.

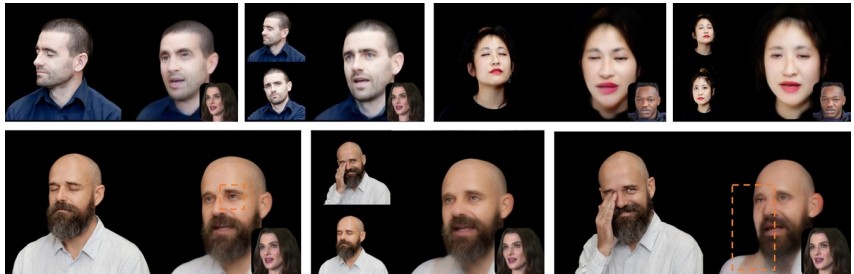

Figure 6: Qualitative results of multi-inputs. In each image, the left side shows input frames, while the right side displays reenactment frames and driving frames. It can be observed that using multiple inputs enhances the performance, especially in cases of closed eyes and occlusions.

Table 3: Ablation results on the VFHQ dataset. Entries in green are the best ones.

| Method | PSNR↑ | SSIM↑ | LPIPS↓ | CSIM↑ | L1↓ | AED↓ | APD↓ | AKD↓ |
|---|---|---|---|---|---|---|---|---|
| w/o PEF | 22.01 | 0.762 | 0.186 | 0.766 | 0.040 | 0.576 | 0.020 | 4.30 |
| w/o global sample | 21.58 | 0.760 | 0.194 | 0.765 | 0.039 | 0.518 | 0.019 | 3.96 |
| point cloud 2000 | 21.95 | 0.761 | 0.193 | 0.750 | 0.040 | 0.497 | 0.020 | 3.86 |
| query 4 points | 22.04 | 0.762 | 0.192 | 0.751 | 0.039 | 0.514 | 0.020 | 3.90 |
| Ours One-in | 22.08 | 0.765 | 0.177 | 0.789 | 0.039 | 0.434 | 0.017 | 3.53 |
| mean Two-in | 22.75 | 0.776 | 0.190 | 0.726 | 0.036 | 0.452 | 0.019 | 3.68 |
| mean Three-in | 23.03 | 0.780 | 0.191 | 0.724 | 0.035 | 0.455 | 0.019 | 3.65 |
| mean Four-in | 23.16 | 0.783 | 0.194 | 0.716 | 0.035 | 0.449 | 0.018 | 3.64 |
| Ours Two-in | 22.86 | 0.779 | 0.169 | 0.771 | 0.035 | 0.411 | 0.017 | 3.44 |
| Ours Three-in | 23.27 | 0.788 | 0.165 | 0.772 | 0.033 | 0.403 | 0.016 | 3.41 |
| Ours Four-in | 23.49 | 0.792 | 0.164 | 0.773 | 0.032 | 0.400 | 0.016 | 3.41 |

## 4.3 ABLATION STUDIES

**Effectiveness of Point-based Expression Field.** To validate the effectiveness of our proposed PEF, we provide FLAME expression parameters directly as a baseline for comparison (w/o PEF in Tab. 3). This method was applied in NeRFace and demonstrated expression control capability. The improvements in AED and AKD clearly indicate that our PEF significantly enhances expression control. We also tried sampling points in a local area with a maximum distance of 1/128 instead of global sampling. The results are shown as w/o global sample in Tab. 3. The results show that our global sampling enhances the details in expressions and the quality of synthesis.

**Effectiveness of Multi Tri-planes Attention.** To validate the effectiveness of our proposed MTA, we established a naive mean-based baseline that averages the tri-planes of multiple images to obtain a merged plane. Table 3 shows the results. We observe that our MTA exhibits better synthesis performance, which we attribute to MTA's ability to avoid feature blurring caused by average fusion.

**Ablation on Hyper-parameters.** We conducted experiments on the selection of hyper-parameters. We randomly selected 2,000 points from 5,023 points of FLAME, and the results in Tab. 3 show that our method can also work on sparse point clouds. This may be attributed to our PEF finding neighboring points from the entire space, which prevents sparse sampling issues. We also reduced the number of query neighbors $K$ from 8 to 4, and the results indicate that our method has some robustness to the number of neighboring points.

## 5 CONCLUSION

In this paper, we have introduced a novel framework for generalizable and precise reconstruction of animatable 3D head avatars. Our approach reconstructs the neural radiance field using only one or a few input images and leverages a point-based expression field to control the expression of synthesized images. Additionally, we have introduced an attention-based fusion module to utilize information from multiple input images. Ablation studies suggest that the proposed Point-based Expression Field (PEF) and Multi Tri-planes Attention (MTA) can enhance synthesis quality and expression control. Our experimental results also demonstrate that our method achieves the most precise expression control and state-of-the-art synthesis quality on multiple benchmark datasets. We believe that our method has a wide range of potential applications due to its strong generalization and precise expression control capabilities.

## 6 ETHICS STATEMENT

Since our framework allows for the reconstruction and reenactment of head avatars, it has a wide range of applications but also carries the potential risk of misuse, such as using it to create fake videos of others, violating privacy, and spreading false information. We are aware of the potential for misuse of our method and strongly discourage such practices. To this end, we have proposed several plans to prevent this technical risk:

- We will add a conspicuous watermark to the synthesized video so that viewers can easily identify whether the video was synthesized by the model. This will significantly reduce the cost for viewers to identify the video and reduce the risk of abuse.

- We limit the identity of the target speaker to virtual identities such as virtual idols, and prohibit the synthesis of real people without formal consent. Furthermore, synthetic videos may only be used for educational or other legitimate purposes (such as online courses) and any misuse will be subject to liability via the tracking methods we present in the next point.

- We will also inject invisible watermarks into the synthesized video to store the IP of the video producer, so that the video producer must consider the potential risks brought by the synthesized video. This will encourage video producers to proactively think about whether their videos will create ethical risks and reduce the possibility of creating abusive videos.

To summarize, as a technology designer, we come up with strict licenses and technologies to prevent abuse of our GPAvatar, a talking face reconstruction system. We think more efforts from governments, society, technology designers, and users are needed to eliminate the abuse of deepfake. Besides, we hope the video maker is aware of the potential risks and responsibilities when using the talking face generation techniques. We believe that, with proper application, our method has the potential to demonstrate significant utility in various real-world scenarios.

## 7 REPRODUCIBILITY STATEMENT

Here we summarize the efforts made to ensure the reproducibility of this work. The model architectures and training details are introduced in Appendix A.1, and we also release the code for the model at https://github.com/xg-chu/GPAvatar. The data processing and evaluation details are introduced in Appendix A.2 and Appendix A.3.

ACKNOWLEDGMENTS

This work was partially supported by JST Moonshot R&D Grant Number JPMJPS2011, CREST Grant Number JPMJCR2015 and Basic Research Grant (Super AI) of Institute for AI and Beyond of the University of Tokyo. This work was also partially supported by JST, the establishment of university fellowships towards the creation of science technology innovation, Grant Number JP-MJFS2108.

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

# A    REPRODUCIBILITY

## A.1    MORE IMPLEMENTATION DETAILS

Specifically, our canonical feature encoder takes the original image of $3 \times 512 \times 512$ as input. We obtain the style code through 4 groups of ResBlock down-sampling, use 3 groups of ResBlock up-sampling to obtain the conditions, and then use StyleGAN (Karras et al., 2019) to output the $3 \times 32 \times 256 \times 256$ tri-planes based on style code and conditions. In the point-based expression field, we assign a 32-dim feature to each point. Since FLAME (Li et al., 2017) contains 5023 points, the total point features size is $5023 \times 32$. During sampling in PEF, we select the nearest 8 points, compute a 39-dim relative position code for each point, and use two fully connected layers to map the 71-dim features to 32-dim. In the multi tri-planes attention module, we employ three linear layers to map the query and key for attention calculation, and the input, output, and hidden layer dimensions are all set to 32. After obtaining 32-dim features from the canonical feature space and expression field, we use three dense layers to map the feature from 64-dim to 128-dim. Subsequently, a single linear layer is employed to predict density, and two linear layers are used to predict RGB values. Finally, we employ a super-resolution network, similar to the encoder, to map the image from $32 \times 128 \times 128$ to $3 \times 512 \times 512$ dimensions. We also provide code for the model in the supplementary materials for reference.

## A.2    MORE DATA PROCESSING DETAILS

We use 8,013 video clips from the VFHQ dataset (Xie et al., 2022) for training, and we uniformly sampled 30 frames from each video clip to ensure that the expressions and poses in each frame were as diverse as possible, resulting in a total of 240,390 frames. We cropped the heads and shoulders from the videos, extracted the 3DMM parameters for each frame (including identity, expression, and camera pose) with (Danecek et al., 2022) and further refined the pose with (Bulat & Tzimiropoulos, 2017). Finally, we resized all these images to $512 \times 512$ pixels.

For the HDTF (Zhang et al., 2021) dataset, we followed the training-testing division in OTAvatar. We conducted a uniform time-based sampling, selecting 100 frames from each of the 19 videos, thereby creating a test split encompassing 1900 frames. As for the VFHQ dataset, we employed a similar approach, uniformly sampling 60 frames from each of the 30 videos. This method ensured that all parts of each test video were sampled as thoroughly as possible.

## A.3    MORE EVALUATION DETAILS

We conducted comparisons with ROME (Khakhulin et al., 2022), StyleHeat (Yin et al., 2022), OTAvatar (Ma et al., 2023), Next3D (Sun et al., 2023), and HideNeRF (Li et al., 2023a) using their official implementations. Since NOFA (Yu et al., 2023b) does not currently provide an official implementation, and GOAvatar (Li et al., 2023b) is a parallel work, it is challenging to make a correct and fair comparison between these two. Additionally, as Next3D (Sun et al., 2023) has not yet provided a formal inversion implementation, we integrated PTI (Roich et al., 2021) for reconstruction within it.

For each method, we utilized the official data pre-processing scripts to obtain their respective input frames, driving frames, and result frames. For all methods, we aligned the facial regions to a uniform size and then resized them to 512x512. It's important to note that the same alignment parameters were applied to both the driving frames and result frames to ensure their correspondence. Subsequently, we computed all metrics on the aligned frames to ensure a fair comparison. Furthermore, as most methods primarily focus on the facial area, our approach actually encompasses a larger region, including parts of the shoulders. During alignment, we used a region closer to the face to make as few modifications as possible to the baseline method's results, which may result in our approach not performing optimally in some metrics.

# B    LIMITATIONS

Our method has some limitations. Specifically, our current FLAME-based model lacks a module to control the shoulders and body, resulting in limited control below the neck Currently, the position of the shoulders in the images generated by our model is generally consistent with the input image.

Additionally, for other areas not modeled by FLAME such as hair and tongue, explicit control is also not feasible. Furthermore, while our aspiration is to achieve real-time reenactment of more than 30 fps, our current performance is pre-real-time for now (approximately 15 fps on the A100 GPU). We leave addressing these limitations for future work.

## C   MORE ABLATION STUDIES

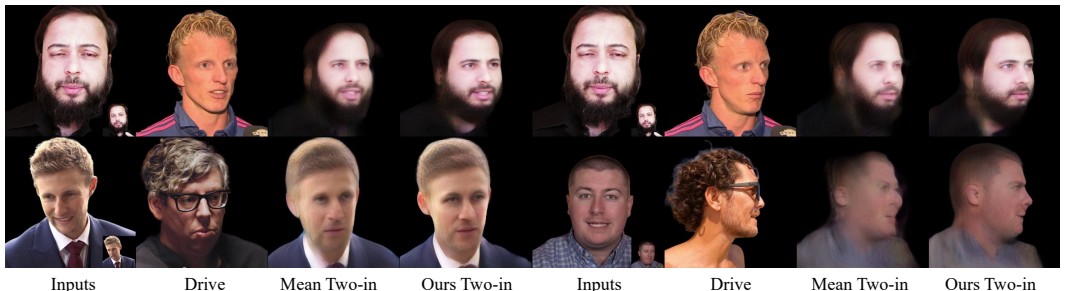

Figure 7:   Qualitative results on VFHQ (Xie et al., 2022) datasets. Compared to the mean baseline, our MTA preserves more details. The smaller of the two inputs is provided to Ours Two-in.

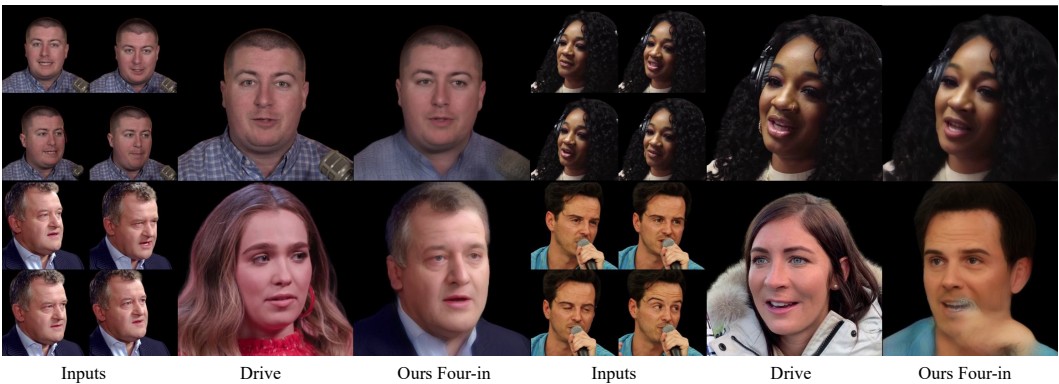

Figure 8:   Qualitative results on VFHQ (Xie et al., 2022) datasets. With four inputs, our method produces sharp and detailed results.

To thoroughly explore design choices for the model, we conducted additional ablation experiments. **More Ablation with MTA.** Fig. 7 presents the qualitative results compared to the naive mean baseline. From the qualitative results, we can observe that the mean baseline excessively smooths the eyes and facial features, leading to a decrease in performance, which aligns with the observations from the quantitative results. Fig. 8 displays the qualitative results of our method when using four images as input. It can be observed that using four images as input does not result in detail smoothing or loss and has a quite good performance.

**Visualization of attention map in MTA.** In order to better evaluate the performance of MTA, we visualized the attention map. As shown in Fig.9, we can see that the model can pay attention to different parts of the face very well. When the input includes left and right faces, the model can provide attention to both sides to achieve a more complete modeling of the whole face. At the same time, the model also pays more attention to the open-eyes input to ensure that the eye area is reconstructed correctly. These visualization results are consistent with our expectations and show the advantages of MTA in fusing multiple image inputs.

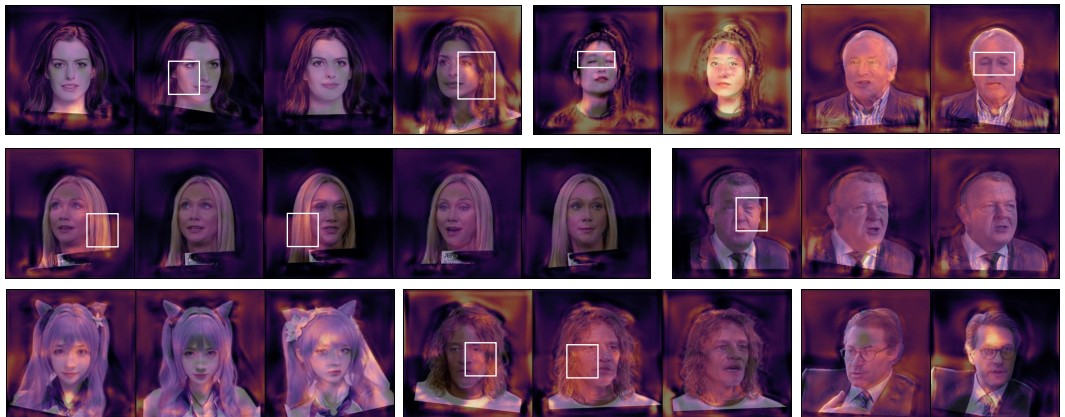

Figure 9: Visualization of attention map in our MTA module. The image is normalized by affine transformation. Red represents higher attention and black represents lower attention. MTA can pay attention to the left and right faces in different inputs, and also pay more attention to the eyes-open photos.

## D   PRELIMINARIES OF FLAME

We use the geometry prior from the FLAME (Li et al., 2017) model, a 3D morphable model known for its geometric accuracy and versatility. It extends beyond static facial models by incorporating expressions, offers precise control over facial features, and is represented parametrically. FLAME finds applications in facial animation, avatar creation, and facial recognition due to its realistic rendering capabilities and flexibility. The FLAME model represents the head shape in the following way:

$$TP(\hat{\beta}, \hat{\theta}, \hat{\psi}) = \bar{T} + BS(\hat{\beta}; S) + BP(\hat{\theta}; P) + BE(\hat{\psi}; E), \tag{6}$$

where $\bar{T}$ is a template mesh, $BS(\hat{\beta}; S)$ is a shape blend-shape function to account for identity-related shape variation, $BP(\hat{\theta}; P)$ is a corrective pose blend-shape to correct pose deformations that cannot be explained solely by linear blend skinning, and expression blend-shapes $BE(\hat{\psi}; E)$ is used to capture facial expressions.

## E   BLENDING MULTIPLE IDENTITIES

We also attempted to synthesize results using different individuals and styles as inputs. As shown in Fig. 10, even in the case of such diverse inputs, our method demonstrated robustness, producing reasonable results while combining features from different persons in the images.

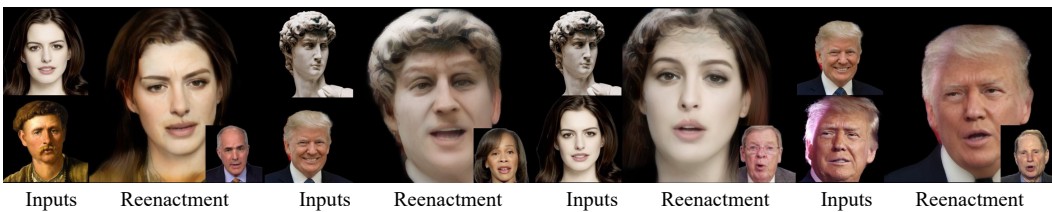

Inputs     Reenactment     Inputs     Reenactment     Inputs     Reenactment     Inputs     Reenactment

Figure 10: Blend results from in-the-wild images. The smaller images are the driving frames.

## F    MORE QUALITATIVE RESULTS

In this section, we showcase more visual results of our method for cross-identity reenactment. Fig. 11 showcases additional results of our method. It's worth noting that, for a fair comparison with other methods, we standardized the evaluation details across all methods, as stated in the evaluation specifics in Appendix A.3. Our method can also encompass more non-facial regions, as illustrated in Fig. 11. Fig. 12 displays the results on VFHQ, and Fig. 13 presents results on HDTF, respectively. We also provide a supplementary video to show more dynamic results.

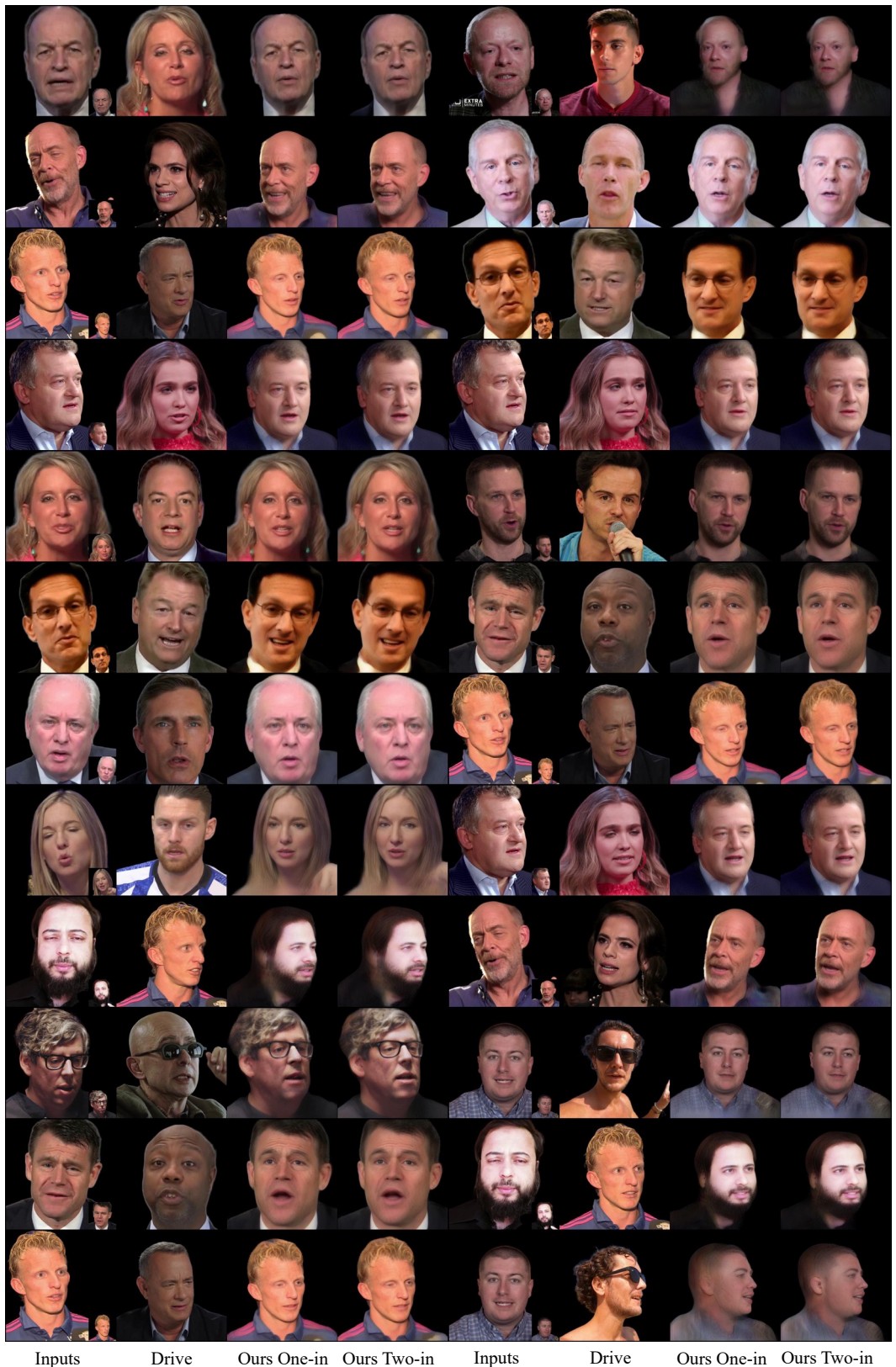

Inputs     Drive    Ours One-in  Ours Two-in    Inputs     Drive   Ours One-in  Ours Two-in

Figure 11: Qualitative results on VFHQ (Xie et al., 2022) datasets. The smaller of the two source inputs is provided to Ours Two-in.

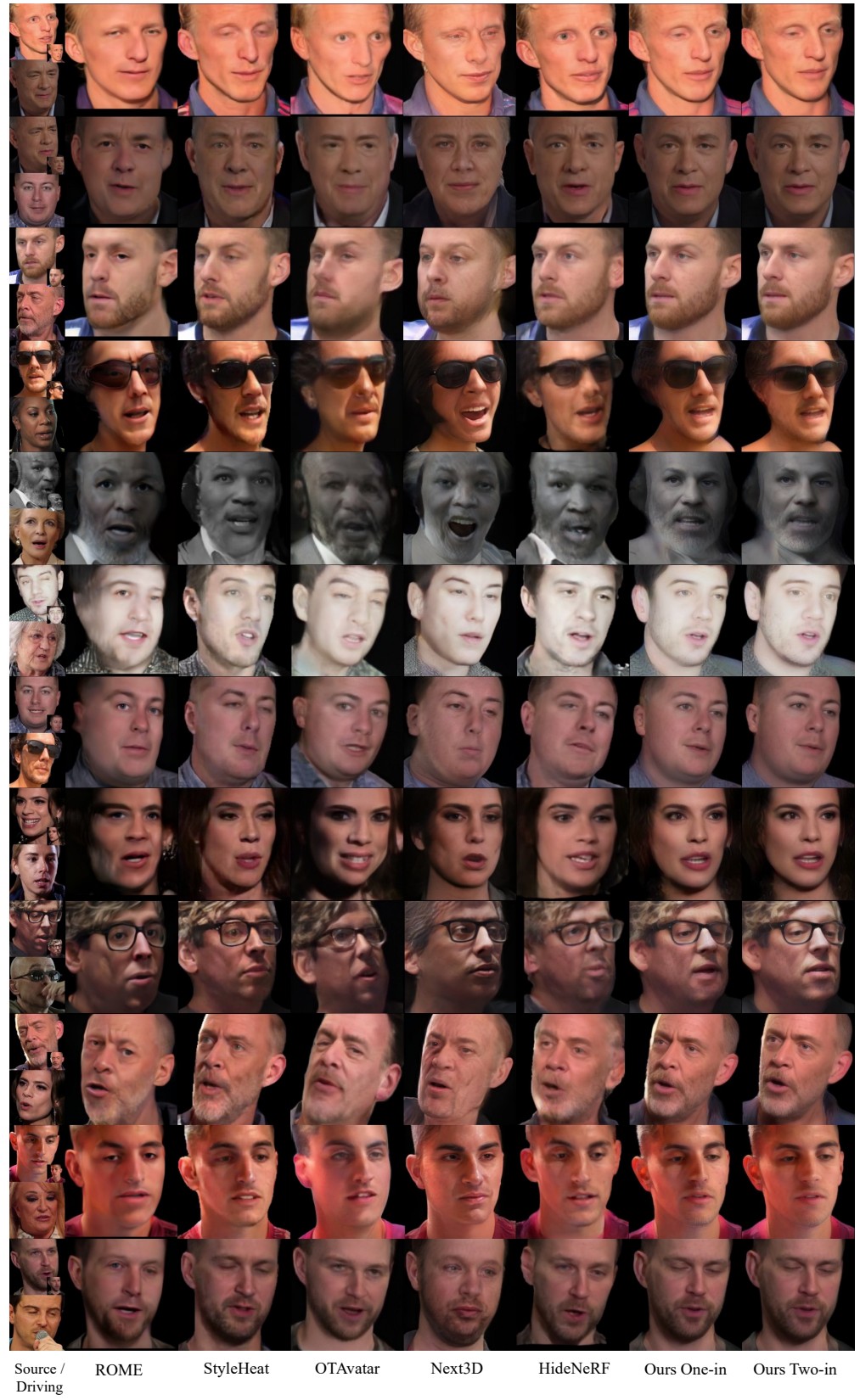

Source / Driving    ROME    StyleHeat    OTAvatar    Next3D    HideNeRF    Ours One-in    Ours Two-in

Figure 12: Qualitative results on VFHQ (Xie et al., 2022) datasets. The smaller of the two source inputs is provided to Ours Two-in.

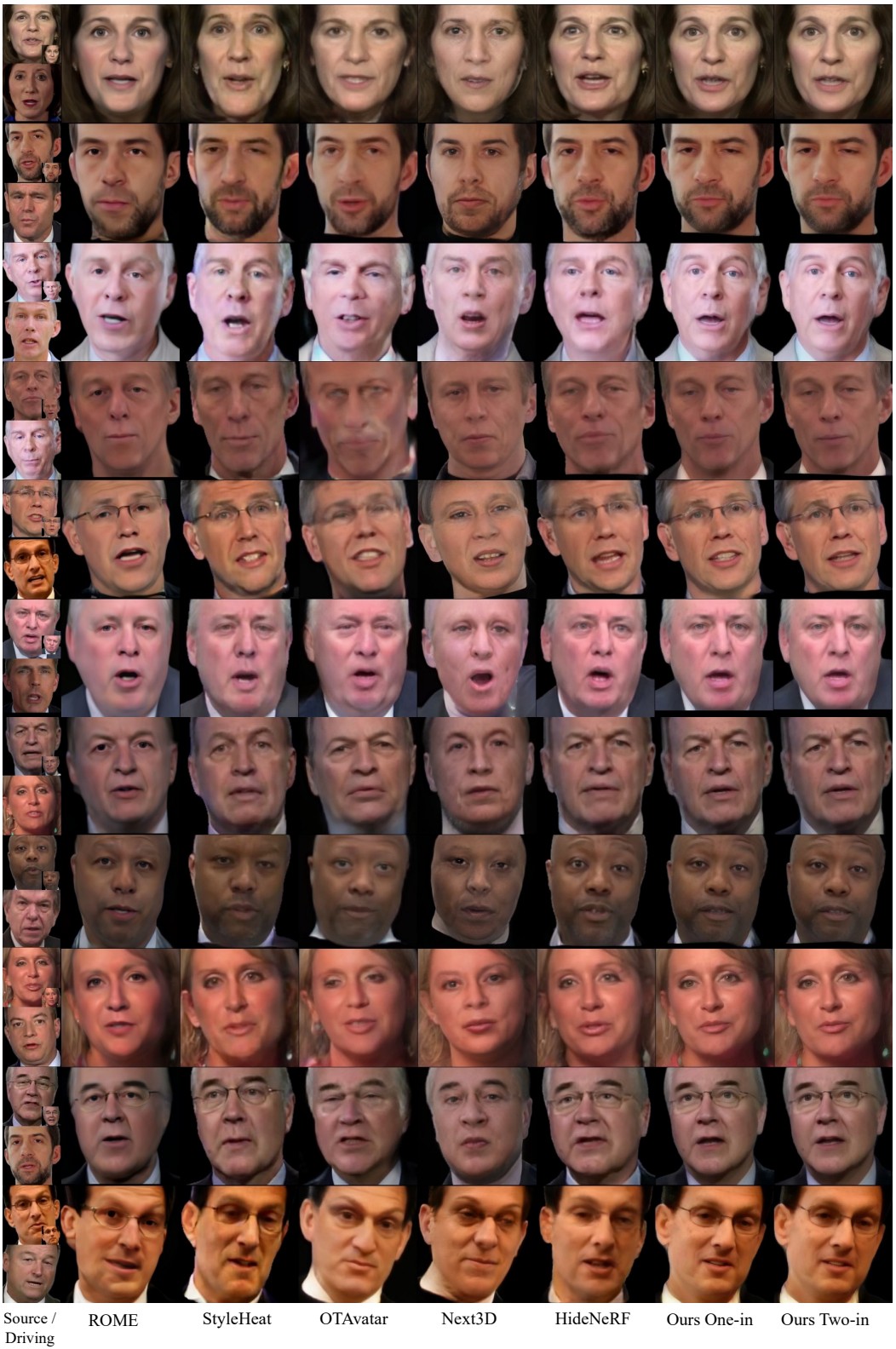

Figure 13: Qualitative results on HDTF (Zhang et al., 2021) datasets. The smaller of the two source inputs is provided to Ours Two-in.

