# OpenReview forum: "GPAvatar: Generalizable and Precise Head Avatar from Image(s)"
_ICLR.cc/2024/Conference — ICLR 2024 poster_

### Official Review · Reviewer_mo8c · 2023-10-30

**Soundness:** 3 good
**Presentation:** 3 good
**Contribution:** 3 good
**Rating:** 6
**Confidence:** 5

**Summary:**

The authors proposed a one-shot/few-shot NeRF-based talking face system. The contributions are twofold: (1) a point-based expression field (PEF) for 3D avatar animation; (2) a multi triplanes attention (MTA) that supports multiple images as input to handle hard cases like occlusion or closed eyes. I like some ideas in this paper. This is a overall well-written paper and is easy to follow. The experiment shows good performance over previous baselines.

However, the identity similarity in the demo video is not as good as previous one-shot 3D talking face methods (such as HiDe-NeRF). Besides, I'm also curious about the performance of this method under large head poses, which is not revealed in the demo.

**Strengths:**

- I like the idea of PEF since it well utilizes the geometry prior of FLAME to help learn the avatar animation in the 3D space.
- Also, it is the first one-shot 3D talking face paper that focuses on the few-shot setting.
- The paper is well-written and is easy to follow.

**Weaknesses:**

- the PEF could well handle the segment modeled by FLAME (such as head and torso), but it cannot handles other parts, such as hair and clothes. See question 1.
- The identity similarity in the demo video is worse than some baseline (HideNeRF).
- The image quality can be improved. For instance, in Figure 1, the predicted images in the second column seems blurry.

**Questions:**

- In  PEF, the expression feature of facial part can be queried from the FLAME mesh, but the non-facial part, such as hair, clothes, and background is not modeled by FLAME. The authors said "we instead search for neighboring points in the entire space", but it is not clear how it bundles the non-facial part in the 3D space with the learnable features.
- The identity similarity in the demo video is not as good as previous one-shot 3D talking face methods (such as HiDe-NeRF), what's the cause?
- The head movement in the provided video is quite gentle. Is there any demo where the head pose is larger (such as side view)? Since one of the biggest advantage of 3D methods over the traditional 2D methods is the good quality under a large view angle, I think this is necessary for the reviewer to assess the performance of a 3D-based work.
- Could you provide the visualization of the attention weights in the multi-reference setting? I'm also curious about the scalability of the MTA, how is the attention map looks like under the two-in/five-in/ten-in ?
- In the demo video, the avatars are driven by audio, it is better to illustrate the way used to obtain the facial expression.

---

> ### Author Response · Authors · 2023-11-20
> **Reply to Reviewer mo8c (Part1)**
>
> Thank you for your constructive feedback and detailed review. We have taken your suggestions and conducted experiments to improve the identity similarity. Our responses below address each of your points.
> ### W1: About the PEF
> Please refer to the Q1.
> ### W2: About the identity similarity
> Please refer to the Q2.
> ### W3:  About the  image quality
> Since our model is only trained on VFHQ data (characters from TV interviews) and inferred on unseen IDs, for some inputs like CG characters, there are some cases where we cannot generate clear reconstruction. We hope that we can solve this problem by enhancing data or improving cross-domain robustness in future work. It is worth noting that even though we have not added these improvements, our model still has amazing generalization ability and is able to handle many cross-domain cases.
> ### Q1: How the PEF handles some non-facial parts.
> Our feature space mainly consists of two parts, the canonical tri-planes feature space and the expression feature space based on the FLAME point cloud (PEF). Non-face areas are overall modeled by the tri-planes space, which is essentially a rigid body. If we limit the feature search of the expression field to the FLAME area, then parts such as hair and torso will be completely modeled as rigid bodies that change as the camera perspective rotates. In order to be able to solve some non-flame modeling areas related to the expressions, we relax the nearest neighbor search distance of the expression feature field, so that some hints of the current expression can be obtained in the entire tri-planes feature space, which makes some non-FLAME areas more flexible. Especially areas such as hair and glasses that are close to the face but are not modeled by FLAME. The quantitative ablation experiment on global sampling also confirmed that this can improve various metrics including reconstruction quality and expression control fineness.
>
> ### Q2: About the identity similarity of our method and HideNeRF
> We examined the technology and implementation used by HideNeRF in detail, and found that Arcface was used to establish identity loss to improve identity consistency. We also adopted the same method and conducted experiments, and we can observe that our identity consistency has also been improved, but at the same time, the reconstruction quality and expression control accuracy have decreased. We believe that we may need to conduct more experiments to find a better balance between multiple losses to obtain better results. We will continue to work on this part. But it is worth noting that the fine control of expressions and the substantial improvement in reconstruction quality are our main contributions and the features we most want to retain.
>
> In the following is the result when we add Arcface-based identity loss L_ID. Among them, ours one-in w/o L_ID is the result in the original paper version, and w L_ID is the result with identity loss added.
> |   Exp on VFHQ              | PSNR↑ |  SSIM↑| LPIPS↓|CSIM↑  |   L1↓ | AED↓  | APD↓  | AKD↓ | CSIM↑ | AED↓  | APD↓  |
> |---                    |---    |---    |---    |---    |---    |---    |---    |---   |---    |---    |---    |
> | HideNeRF              | 20.07 | 0.745 | 0.204 | 0.794 | 0.056 | 0.521 | 0.031 | 5.33 | 0.558 | 1.024 | 0.044 |
> | Ours One-in w/o L_ID  | **22.08** | **0.765** | **0.177** | 0.789 | **0.039** | **0.434** | **0.017**| **3.53** | 0.558 | **0.910** | **0.034** |
> | Ours One-in w L_ID    | 21.48 | 0.759 | 0.187 | **0.797** | 0.043 | 0.464 | 0.017 | 3.84 | **0.574** | 0.932 | 0.035 |
>
> |     Exp on HDTF         | PSNR↑ |  SSIM↑| LPIPS↓|CSIM↑  |   L1↓ | AED↓  | APD↓  | AKD↓ | CSIM↑ | AED↓  | APD↓  |
> |---                    |---    |---    |---    |---    |---    |---    |---    |---   |---    |---    |---    |
> | HideNeRF              | 21.38 | 0.803 | 0.147 | 0.907 | 0.038 | 0.499 | 0.027 | 4.33 | 0.803 | 1.031 | 0.032 |
> | Ours One-in w/o L_ID  | **24.21** | **0.834** | **0.131** | 0.871 | **0.029** | **0.427** | **0.012** | **3.06**| 0.790 | **0.869** | **0.020** |
> | Ours One-in w L_ID    | 23.67 | 0.821 | 0.136 | **0.910** | 0.034 | 0.443 | 0.013 | 3.45 | **0.812** | 0.892 | 0.020 |

---

> ### Author Response · Authors · 2023-11-20
> **Reply to Reviewer mo8c (Part2)**
>
> ### Q3: About the larger head pose
> We agreed that a large change in the view angle is crucial for evaluating a 3D-based method. The animation sequence in our audio-driven demonstration is extracted from SadTalker, which always has smooth and gentle head movements. But in our paper numerous results show significant head movements, even in side views (as illustrated in Figure 7). These results show that our method effectively inherits the advantages of 3D methods in view angle changes.
>
> ### Q4: The visualization of the attention weights
> We appreciate the reviewer's insightful suggestion to show attention visualizations, which can better show how our MTA module works. For now, we have included these visualizations in the revised version (Sec.C and Figure.9). These attention visualization results are very interesting and show that our method can indeed pay attention to and combine different angles of people in different images.
>
> ### Q5: How we obtain facial expression from audio
> We employ a widely used audio-driven motion generator: SadTalker to generate head motion based on audio input, although other similar methods can also be used. SadTalker uses an image and a piece of audio to generate a video, and then we track the 3DMM parameters in that video. We then use these parameters to animate the GPAvatar. We will add descriptions in future demos.

---

> > ### Comment · Reviewer_mo8c · 2023-11-22
> > **Reply to Rebuttal**
> >
> > Thanks for the reply. The authors have addressed most of my concerns, so I keep my initial rating of 6. By the way, do you have the plan to open-source the code in the future?

---

> > > ### Author Response · Authors · 2023-11-22
> > > **Reply to Reviewer mo8c**
> > >
> > > Thanks for your kind reply and comments. Of course, we will open source the code in the future.

---

### Official Review · Reviewer_q9Mp · 2023-11-01

**Soundness:** 3 good
**Presentation:** 2 fair
**Contribution:** 2 fair
**Rating:** 6
**Confidence:** 4

**Summary:**

This paper proposes a method that reconstructs 3D head avatars from images and synthesizes realistic talking head videos. It takes as input a single or a small number of face images and reconstructs 3D head avatars in a single forward pass. It extends the formulation of NERFS and proposes a dynamic point-based expression field that is driven by a point cloud, motivated by the need to have an accurate control of facial expressions. In addition, the proposed method adopts a Multi Tri-planes Attention (MTA) fusion module that facilitates the 3D representation of the scene and the incorporation of information from multiple input images. The proposed method is compared with several SOTA methods and achieves promising results.

**Strengths:**

+ The proposed method achieves promising results and the supplementary videos show that the videos synthesized by the proposed method are in general realistic and visually pleasing.

+ The experimental evaluation is detailed and systematic. The proposed method is compared with several recent SOTA methods that solve the same problem.

**Weaknesses:**

- The presentation in several parts of the paper, especially in the methodology, is unclear and needs several clarifications. See detailed comments in Questions below.

- The following paper is not cited, despite the fact that it is very closely related in terms of methodology:

Athar, S., Shu, Z. and Samaras, D., 2023, January. Flame-in-nerf: Neural control of radiance fields for free view face animation. In 2023 IEEE 17th International Conference on Automatic Face and Gesture Recognition (FG) (pp. 1-8). IEEE.

The above un-cited paper also uses a FLAME-based representation of the 3D face and extends the formulation of NERFS to achieve realistic face animation with expression control. The similarities and differences with the proposed method are not discussed and the Flame-in-nerf is not included in the comparisons of the experimental section.  This raises concerns in terms of the real novelty and contributions of the proposed method.

- Furthermore, there are also several other closely related works that are not cited. For example:

Jiaxiang Tang, Kaisiyuan Wang, Hang Zhou, Xiaokang Chen, Dongliang He, Tianshu Hu, Jingtuo Liu, Gang Zeng, and Jingdong Wang. Real-time neural radiance talking portrait synthesis via audio-spatial decomposition. arXiv preprint arXiv:2211.12368, 2022.

Yudong Guo, Keyu Chen, Sen Liang, Yong-Jin Liu, Hujun Bao, and Juyong Zhang. Ad-nerf: Audio driven neural radiance fields for talking head synthesis. In ICCV, pp. 5784–5794, 2021.

Yu, H., Niinuma, K. and Jeni, L.A., 2023, January. CoNFies: Controllable Neural Face Avatars. In 2023 IEEE 17th International Conference on Automatic Face and Gesture Recognition (FG) (pp. 1-8). IEEE.

**Questions:**

- Section 3.2: the paper fails to clearly explain how the expression information affects the process of building a point-based expression field.

- Section 3.3: the paper provides insufficient details about about how the canonical encoder is defined and built.

- Figure 4: For several columns with results, it is unclear which is the corresponding method. There is one column more than the number of methods in the caption and one column more than the columns of Figure 5. This creates confusions.

- Equation (1): the definition of w_i does not seem to make sense. Is there a missing norm in the denominator?

- After Equation (2): N is refereed to as the "input number", but apparently it should be referred to as the number of input images

**Details Of Ethics Concerns:**

As is the case for all methods of this field, the proposed method could be misused in order to create deep fake videos of a person without their consent. This raises issues of misinformation, privacy and security. Section 6 includes some discussion about this issues. However, this discussion could have been more detailed, with a more in-depth discussion of specific mitigation measures.

---

> ### Author Response · Authors · 2023-11-20
> **Reply to Reviewer q9Mp**
>
> Thank you for your constructive feedback and time to review our paper. We have taken your suggestions and further refined our paper. Please find our detailed responses to each of your points raised below:
> ### W1: Methodology needs several clarifications
> Please refer to the responses to the corresponding questions.
> ### W2-3: About the missing citations.
> Thanks for pointing this out. We acknowledge the relevance of this paper and include it in our updated related work section.
> However, it is worth noting that our method is not comparable to these methods. Our approach focuses on oneshot and fewshot settings, meaning the input is only one or a few images, no training on new identities is required, and our model is driven by motion sequences based on FLAME parameters.
> Among these papers, the paper [1] requires videos as input (may have thousands of images), and each ID-specific model needs to be trained based on the videos. This prevents the model from making inferences on unseen IDs, but requires data from that ID for training. This is different from our setting, where we use unseen IDs during inference to evaluate our method, which is not possible in paper [1]. At the same time, the way it uses FLAME is also different from us. Our motivation is to avoid the loss of expression information, but the paper [1] adopts a method similar to Figure.2(c), by directly inputting the expression vector to control the expression, which will obviously lead to expression Loss of information.
>
> Papers [2] and [3] use audio signals as the driving method, which makes it difficult to accurately control the avatar to make the desired expression, and since we use FLAME parameters to control the avatar to make expressions, the difference in control signals makes we differ from papers [2] and [3] and cannot evaluate in the same setting.
>
> The setting of paper [4] is similar to paper [1], requiring the use of video as training data and cannot infer with an unseen ID. This makes paper [4] not comparable to our method.
>
> ### Q1: Sec 3.2, how the expression information affects the PEF?
> Our feature space mainly consists of two parts, the canonical tri-planes feature space and the expression feature space based on FLAME point cloud (PEF). Expression information is given in the form of FLAME parameters, and the FLAME model dynamically forms a point cloud based on these parameters. That is, different FLAME parameters will generate point clouds with different positions. Since our PEF is based on these point clouds, the feature queried from PEF will also be changed with different expression parameters.
>
> ### Q2: Sec 3.3, how do we build the canonical encoder?
> We adopt the same Style-UNet structure as in GFPGAN with only modification of the input and output size.
> In short，we obtain the style code through 4 groups of ResBlock down-sampling, use 3 groups of ResBlock up-sampling to obtain the conditions, and then use StyleGAN to output the 3 × 32 × 256 × 256 tri-planes based on style code and conditions. For more implementration details please refer to Sec.A.1, and we will also release the code and checkpoints to facilitate reproducibility and further research.
>
> ### Q3: Figure 4, missing column title
> The last two columns actually correspond to "Ours One-in" and "Ours Two-in". For a fair comparison, we decided to only compare here the driving results of a single image as input. In the revised version we have removed "Ours Two-in" in Figure 4. Results of Ours Two-in or more can be referenced from other Figures (like Figure 6,7,8,9).
>
> ### Q4-Q5: About the equations.
> We appreciate the reviewers for identifying sign errors in our formulas. These mistakes have been rectified in the revised version. The corrected equation is $$f_{exp,x} = \sum_{i}^{K} \frac{w_i}{\sum_{j}^{K} w_j} L_p(f_{i}, F_{pos}(p_i-x)), \text{where}~w_i=\frac{1}{p_i-x},$$ where $x$ is the coordinate to be queried, $K$ is the number of neighbors, $f_i$ is the corresponding feature, $p_i$ is the point coordinate, $L_p$ is the linear layers and $F_{pos}$ is the frequency positional encoding function. In this equation, the $w_i$ ensures that the nearest point has the highest weight when contributing to the feature.
>
> In Equation (2): $N$ should be referred to the number of input images and we also correct it in the revised version.
>
> ### Ethic concerns:
> We agree with you that there are potential ethical risks with talking head generation. Please refer to the "Response to Ethics Concerns" for details. We have also added more discussion about ethics in Sec.7 of the revised paper.

---

### Official Review · Reviewer_MrBB · 2023-11-05

**Soundness:** 3 good
**Presentation:** 3 good
**Contribution:** 3 good
**Rating:** 6
**Confidence:** 4

**Summary:**

In this paper, a novel framework named GPAvatar is introduced, designed to reconstruct 3D head avatars seamlessly from one or multiple images in a single forward pass. The key novelty lies in the incorporation of a dynamic point-based expression field, guided by a point cloud, to intricately and efficiently capture facial expressions. The authors present the concept of a dynamic Point-based Expression Field (PEF), enabling accuracy and control of expressions across different identities. Additionally, they introduce a Multi Tri-planes Attention (MTA) fusion module, capable of handling a varied number of input images with precision.

**Strengths:**

- Overall, the paper is well-organized and easy to follow. The motivation is clear. The figures and tables are informative.

- Experimental results demonstrate that the proposed method achieves the most precise expression control and state-of-the-art synthesis quality (StyleHeat, ROME, OTAvatar, and Next3D) (based on NeRF and 3D generative models)n on multiple on VFHQ and HDTF benchmark datasets.

**Weaknesses:**

- The model proposed has overall more trainable parameters compared to baseline models, which could potentially bring in some unfairness during comparison with other works.
- No discussion about the limitations of the approach?

**Questions:**

- It is not clear to me how the model captures the subtle information such as closed eyes ?
- why the normalization of the weight wi in the equation is required?
- Several terms in the equation (page 3) I_t= R (....) are not defined. What R means?

**Details Of Ethics Concerns:**

Since the proposed framework allows of head avatars, it has a wide range of applications but also carries the potential risk of misuse. The authors are also considering methods like adding watermarks to synthetic videos to prevent misuse.

---

> ### Author Response · Authors · 2023-11-20
> **Response to Reviewer MrBB**
>
> Thank you for your kind feedback and your pointers and questions. We have refined our revised version paper based on your suggestions. Please find our answers for each of your points below.
> ### W1. About trainable parameters
> Given the wide range of parameters in these models, there is a concern regarding fair comparisons. However, it is quite difficult to evaluate these methods at the same parameter level due to different custom structures. Notably, our method does not have the most trainable parameter among all the baselines: our model has 96M parameters but HideNeRF has 569 million parameters.
> ### W2. About the limitations of our approach
> We appreciate the reviewer for pointing out this missing part. Specifically, our current FLAME-based model lacks a module to control the shoulders and body, resulting in limited control below the neck (the shoulder position generally aligns with the input image for now). Additionally, for other areas not modeled by FLAME, such as hair and tongue, explicit control is not feasible. Furthermore, while our aspiration is to achieve real-time reenactment of more than 30 fps, our current performance is pre-real-time for now (approximately 15 fps on the A100 GPU). We added the discussion about limitations in our revised paper (Sec.B), and will solve these problems as future work.
> ### Q1. About how we captures the subtle information
> We avoid the information loss resulting from over-processing using our PEF (illustrated in Figure 2). Consequently, any subtle details captured by FLAME directly influence the point positions in the PEF, thereby contributing to the rendering results.
> ### Q2-Q3. About the normalization weights in equation 1 and R in equation (Page 3).
> We appreciate the reviewers for identifying sign errors in our formulas. These mistakes have been rectified in the revised version. The corrected equation is
> $$f_{exp,x} = \sum_{i}^{K} \frac{w_i}{\sum_{j}^{K} w_j} L_p(f_{i}, F_{pos}(p_i-x)), \text{where}~w_i=\frac{1}{p_i-x}, $$ where $x$ is the coordinate to be queried, $K$ is the number of neighbors, $f_i$ is the corresponding feature, $p_i$ is the point coordinate, $L_p$ is the linear layers and $F_{pos}$ is the frequency positional encoding function. In this equation,  the  $w_i$ ensures that the nearest point has the highest weight when contributing to the feature. While for the $R$ in equation (page 3), it refers to the volume rendering process in NeRF, which renders the merged feature space into the result image.

---

### Author Response · Authors · 2023-11-20
**Common response**

***
We thank the reviewers for their insightful comments and acknowledging that the paper is well organized (Mrbb/mo8c), well-motivated (Mrbb) and easy to follow (Mrbb/mo8c); the experiments are detailed and achieve good results( Mrbb and q9mp ), likes to our idea and found out we were the first one-shot 3D talking face paper that focuses on the few-shot setting (mo8c).
We have carefully considered your comments and will take them into account to further improve the quality of our work. We have updated our PDF accordingly and listed the modifications to the pdf here:
 - We improve the description for canonical encoder, point-based expression field in Sec 3.1, 3.2. We also correct the Figure 4.
 - We correct the mistakes in the equation (page 3) and equation (1).
 - We add a discussion about limitations in the Sec.B. Meanwhile we improve and talk deeper for ethics concerns in Sec.7.
 - We add a visualization of attention maps in Sec.C and Figure 9.
 - We add missing citations of related works in Sec2.1.
Please also find below our responses to specific concerns of each individual reviewer. We remain committed to addressing any further questions or concerns from the reviewers promptly.

Best regards,

The authors

---

### Author Response · Authors · 2023-11-20
**Response to Ethics Concerns**

We would like to thank Reviewer MrBB and q9Mp for their constructive reviews of the ethics concerns. Here we hope our response could address your concerns and we also add a more in-depth discussion of ethics in Sec.7 of the revised paper.

We do agree with you that talking head generation makes it easier to produce "deepfakes" and that there are potential ethical risks that are common to all head generation related submissions. To this end, we have proposed several plans to prevent this technical risk:

 - We limit the identity of the target speaker to virtual identities such as virtual idols, and prohibit the synthesis of real people without formal consent. Furthermore, synthetic videos may only be used for educational or other legitimate purposes (such as online courses) and any misuse will be subject to liability via the tracking methods we present in the next point.
 - We will add a conspicuous watermark to the synthesized video so that viewers can easily identify whether the video was synthesized by the model.
 - We will also inject invisible watermarks into the synthesized video to store the IP or machine ID of the video producer, so that the video producer must consider the potential risks brought by the synthesized video.

In summary, as technical designers, we propose strict permissions and white-box detectors to prevent misuse of the speaking face generation system. We believe that more efforts are needed to collaborate among governments, society, technology designers, and model users to eliminate the abuse of deepfakes.

---

### Meta-Review · Area_Chair_i4zm · 2023-12-22

**Metareview:**

The paper introduces GPAvatar, a framework for 3D head avatars reconstruction from one or several images in a single forward pass. The main technical novelty is in capturing precise facial expressions via incorporation of a dynamic point-based expression field, guided by a point cloud.

All reviewers are on a positive side about this paper, and liked the main idea of point-based expression field as well as the empirical results. In the initial reviews there were some questions regarding 1) missing references and baselines, 2) Eq (1), 3) identity similarity among others, and the rebuttal was able to address most of them. The final decision is to accept this paper. Nonetheless we urge the authors to address as many concerns as possible in the final version.

**Justification For Why Not Higher Score:**

All reviewers agree with acceptance, but not strong acceptance for this work.

**Justification For Why Not Lower Score:**

All reviewers and AC are positive about this work.

---

### Decision · Program_Chairs · 2024-01-16

Accept (poster)